# Community Health Survey of Residents Living Near a Solid Waste Open Dumpsite in Sabak, Kelantan, Malaysia

**DOI:** 10.3390/ijerph17010311

**Published:** 2020-01-02

**Authors:** Bachok Norsa’adah, Omar Salinah, Nyi Nyi Naing, Abdullah Sarimah

**Affiliations:** 1Unit of Biostatistics and Research Methodology, School of Medical Sciences, Universiti Sains Malaysia, Kubang Kerian 16150, Kelantan, Malaysia; sarimah@usm.my; 2Disease Control and Epidemiology Branch, Melaka Tengah Health Office, Melaka 75150, Malaysia; drsalinah@gmail.com; 3Faculty of Medicine, Medical Campus, Universiti Sultan Zainal Abidin, Kuala Terengganu 20400, Terengganu, Malaysia; syedhatim@unisza.edu.my

**Keywords:** dumpsite, landfill, dumping waste, solid waste, environmental health, Malaysia

## Abstract

The management of waste materials is a serious problem worldwide, especially in urbanizing countries like Malaysia. This study was conducted to compare the prevalence of health symptoms and diseases diagnosed among residents exposed to the solid waste open dumpsite in the suburb of Sabak with the non-exposed community. Research related to exposure to solid waste dumping with complete health problems has never been combined in one study. A comparative cross-sectional study was conducted. The exposed group included residents within a 1 km radius and the non-exposed group included residents between a 2.5 and 4.0 km radius from the dumpsite. The selected residents were interviewed using validated, structured questionnaires. A total of 170 residents from the exposed group and 119 residents from the non-exposed group were selected. The mean (SD) duration time of residence was 22.6 (18.9) years for the exposed group and 15.0 (12.0) years for the non-exposed group. Dumpsite exposure was significantly associated with sore throat (adjusted odd ratio (AOR) 1.88; 95% confidence interval (CI): 1.05, 3.38; *p* = 0.031), diabetes mellitus (AOR 2.84; 95% CI: 1.10, 7.30; *p* = 0.021) and hypertension (AOR 2.56; 95% CI: 1.27, 5.13; *p* = 0.006). This study provides evidence that the unsanitary solid waste disposal in Malaysia is hazardous to the health of residents in the surrounding 1 km, and efforts are needed to minimize the hazards.

## 1. Introduction

Urbanization, economic development, and a rapidly growing population result in massive quantities of waste materials requiring proper management [1]. Worldwide, the management of waste materials is a serious problem, especially in developing countries, as it is expensive to design, maintain, and implement. Malaysia, with a population of 32 million, is facing an increase of generations, leading to large accumulation of waste. Kuala Lumpur, the capital city of Malaysia, is expected to produce more than 2.8 million tons of solid waste in 2020 [2]. While moving forward to accomplish the goal of an industrialized country, Malaysia is facing serious solid waste management challenges [3,4].

Open dumping is practiced in most cases [5] and occurs in approximately 50% of the total landfills in Malaysia [6]. More than 230 landfills have been reported in Malaysia and most of them are crude dumping grounds [6]. It is inevitable that the amount of land available will become scarce for providing space for solid waste disposal. This development leads to substantial social, economic, and environmental problems, especially in the crises of land usage. Landfills cause pollution of natural resources and many environmental problems such as health hazards and contamination of surface water and groundwater [4,6,7].

Leachate is produced when water filters downward through a landfill and picks up dissolved materials from the decomposing wastes which can contaminate the groundwater and surface water, which are the sources of drinking water [2,5,8,9]. It is comprised of organic and inorganic pollutants, which include phenols, toluene, benzene, ammonia, dioxins, polychlorinated biphenyls, chlorinated pesticides, heavy metals, and endocrine-disrupting chemicals. This contamination may enter the food chain and endanger public health. A study in India carried out an analysis of the impact of landfills on solid water, leachate, and groundwater by comparing the hydrochemical natures [10]. The result showed that the samples had a high measurement of heavy metals. The groundwater sample showed high contamination of leachate as indicated by the potassium/magnesium ratio.

Many chemicals known to have harmful effects on human health are potentially present in landfill sites [5,8,11]. Exposure to a landfill is associated with health problems such as respiratory symptoms; irritation of the skin, nose and eyes; gastrointestinal problems; fatigue; headache; psychological disorders; and allergies [8]. The wastes may contain chemicals that can cause health risks like cancer, birth defects, preterm babies, and congenital disorders. The excess risk of congenital anomalies and low birth weight for the population living within 2 km of landfills was 2% and 6%, respectively [12].

Odor-producing chemicals such as hydrogen sulphide and ammonia can cause acute effects such as nausea, fatigue, headache, and irritation of the eyes, nose, and throat [13,14]. Pathogenic organisms from solid waste such as bacteria, yeasts, protozoa, worms, and viruses in the landfill may cause diseases to the exposed individuals. Biological vectors like insects and rodents may directly or indirectly transmit disease agents from solid waste to humans [11].

Gases may be absorbed through underground soil and contaminate nearby plants, animals, and humans [13]. Samples of surface soil from open waste dumping sites in Islamabad City have shown high concentrations of pH, total dissolved solids, electrical conductivity, and heavy metals in comparison to control sites [15]. A study conducted in Kolkata reported that the soils of a century-old landfill site was heavily contaminated with heavy metals and their water-soluble forms that are potentially toxic [16]. This is detrimental to microbial biomass and soils activities. This was supported by a study that reported fewer plant species at the disposal sites compared to a control area, which was attributed to changes in soil characteristics of the waste disposal sites [15].

Our literature search revealed a limited study on this issue in recent times, especially in developing countries like Malaysia. The literature has also reported limited evidence of an association between waste processing (mainly landfills and incineration) and health effects [12]. Therefore, this study aimed to determine an association between landfill exposure and the health of nearby residents. We would like to establish epidemiological evidence of a potential human health effect of solid waste landfills. This study would include health symptoms, illnesses, diseases, and maternal and child health of the residents. Research linking solid waste dumping exposure with health symptoms, chronic diseases, and child and maternal health has never been collected together in one study. The results of this study would provide information to the relevant authorities so that they can decide on the regulation of landfill sites and increase knowledge of the need for proper management of landfills to safeguard the health of the residents living nearby. The findings of this study can also serve as the basis for future research on landfills’ impact on health in Malaysia, which is seriously lacking.

## 2. Materials and Methods

A comparative cross-sectional study was conducted in which the sampling frame was stratified into two zones: exposed and non-exposed areas according to the World Health Organization (WHO) criteria [17]. The zones were mapped using an electronic map from the Department of Survey and Mapping Malaysia, which displays the location and distance of the houses in each village from the Sabak dumpsite.

The exposed subjects were residents within a 1 km radius of the dumpsite [17]. The two villages that surround the dumpsite were included. The population data of residents in those villages were provided by the Health Department. The data showed that village A had 223 houses with 976 residents while village B had 260 houses with 1281 residents. Approximately 483 inhabited households were in the defined area with a radius of 1 km. The non-exposed subjects were residents between a 2.5 and 4.0 km radius from the dumpsite. One village was selected for the non-exposed area which had 371 houses with 1760 residents. The village is right next to the exposed villages and chosen because we would like to control the similarity of other environmental exposures other than the dumpsite, such as industrial activities, roads, and motor vehicles. The maps of the dumpsite and selected villages provided by the Kelantan Department of Survey and Mapping are shown in Figure 1 and Figure 2.

For both groups, houses in the identified area and household members who fulfilled the inclusion and exclusion criteria were randomly selected. We used systematic sampling to select second consecutive homes in each village. The eligibility criteria were as follows: (i) permanent resident of the area for at least one year and after the opening of the dumpsite, (ii) age 20 years and above, and (iii) head of the family or the next of kin. Residents were excluded if they had a mental problem, were not at home during the visits, or had resided in a dumpsite area prior to residing in the control area.

### 2.1. Area Description and Study Population

The Sabak dumpsite is located about 13 km from Kota Bharu, the capital city of Kelantan State in Malaysia, at 6°10′31.2″ northern latitude and 102°18′43.4″ eastern longitude. It began operation in September 1987 and covered more than 22 hectares. Approximately 200–230 tons of municipal solid waste were added daily in 2003. It has been reported that 50% to 60% of municipal solid waste in Malaysia is organic waste, including food waste [2]. Mainly domestic, commercial, construction/demolition debris, agricultural waste, non-hazardous sludge from municipal sewage treatment facilities, and non-toxic industrial waste from the Kota Bharu area were dumped in this dumpsite. Kota Bharu is the most populous district in the state of Kelantan.

In earlier operations, the wastes were deposited into dug holes, which were then filled up. The method of waste disposal in this area is called control tipping, where the solid waste is buried in sections and later covered with soil. This minimizes the unsightly appearance, foul smell, and problems with insects and rodents that commonly exist in open dumpsites. However, this method implements only limited measures of standards for sanitary landfill. It does not control the adverse environmental impacts of landfills, such as contamination of groundwater by leachate and emissions of landfill gas (LFG). Since 2001, the waste has been openly deposited and not filled.

The Sabak dumpsite has a clay type of soil which may provide natural purification of the leachate by means of ion exchange, filtration, absorption, precipitation, and biodegradation to minimize some of the leachate hazards. There are no engineered techniques for leachate and LFG management and no monitoring by the Department of Environment for groundwater quality or LFG emission.

### 2.2. Data Collection

A questionnaire was developed as a result of discussion among researchers, officials from the Kota Bharu Municipal Council, the Pengkalan Chepa Health Office, and the literature review [8,12,17,18,19]. The questionnaires were validated in a pre-test (Appendix A). They were divided into four sections. Section A consisted of socio-demographic details, including age, sex, race, education, occupation, household income, smoking status, and length of residence. Section B was on health symptoms among family members in the past month prior to the survey, such as respiratory, skin, and gastrointestinal symptoms. Section C involved self-reported diseases such as cancer, hypertension, diabetes mellitus, heart disease, tuberculosis, asthma, pneumonia, typhoid fever, and cholera that were experience or currently suffered by the residents since residing in the area for at least a year and occurred after the opening of the dumpsite in September 1987. These diseases should be diagnosed by a doctor and reconfirmed by checking the subject’s admission or medical card. The month and year of onset of each disease and any hospital admission were also collected. Section D was about the reproductive history of the women, which includes the number of children born in the past 10 years of the study, birth weight, age of mother at each pregnancy, any obstetric or medical complication, congenital abnormality if present and the type, and abortion occurring within the above time period. Similarly, all obstetric histories were confirmed by the available antenatal cards. The type of congenital abnormality was also rechecked via medical card. The reproductive outcome was considered when occurring after at least one year of maternal residency in the zone.

The residents were recruited through door-to-door interviews. The purpose of the study was explained to the residents and they were invited to participate voluntarily. The written consents were signed, and the respondents were interviewed using the guided questionnaires by our three enumerators. The survey took between 30 and 45 min to complete. The researchers also set up appointments at households within the defined study area to improve the efficiency of the data collection process. The researchers attempted to contact each household twice. This study was ethically approved by the Human Ethics Committee of our institution. The study was carried out in accordance with the rules of the 1975 Declaration of Helsinki.

### 2.3. Data Analysis

Statistical analysis was carried out using SPSS version 24 (IBM, Armonk, NY, USA). Descriptive statistics were analyzed for the socio-demographic information of the respondents, respondents’ health-related problems, disease prevalence among family members, and reproductive health of female residents. The Pearson chi-square test or the Fisher’s exact test were used for comparing proportions between two or more independent groups. Meanwhile, the t-test was used to compare the means difference of the numerical variables between respondents of the exposed and non-exposed areas.

Multiple logistic regression analysis was applied to determine the association between dumpsite exposure and disease, symptoms, or conditions outcome. The dumpsite exposure was the independent variable. The forward and backward stepwise procedures were used for variable selection. The final model was selected that includes dumpsite exposure, the best fit, and the simplest model possible, describing the association between the dumpsite exposure and other independent variables and the outcomes such as health symptoms, diseases, or reproductive health. The other independent variables were age, sex, race, education, occupation, household income, smoking status, factory exposure, water supply, growing own vegetables, rearing own chicken, distance from the landfill, and length of residence.

The model fitness was assessed with the Hosmer–Lemeshow test. Results were presented with crude and adjusted odds ratio (OR), 95% confidence interval (CI) and *p*-value. The level of statistical significance was set at *p* value less than 0.05.

## 3. Results

The socio-demographic details of the exposed group and non-exposed group are shown in Table 1. The mean (SD) age was 46.3 (13.5) years for the exposed group and 38.1 (9.8) years for the non-exposed group. The majority of respondents were female (76.5%) and housewives (53.6%). The mean (SD) duration of residence was 22.6 (18.9) years for the exposed group and 15.0 (12.0) years for the non-exposed group. The mean (SD) distance of the residence from the Sabak dumpsite was 0.45 (0.24) km and 3.06 (0.37) km for the exposed and non-exposed group, respectively.

There were significant differences between groups in age, sex, education level, working in factory, water supply, growing own vegetables, duration of residence, and distance from the dumpsite to the house. The exposed group was significantly older, had more men, less education, less working in factory, used more water supply from Kelantan Water Company (Kelanta, Malaysia), and fewer grown vegetables than the non-exposed group. The exposed group resided significantly longer in the area compared to the non-exposed group.

### 3.1. Comparison of Self-Reported Health Symptoms in Past One Month between Exposed and Non-Exposed Residents

Table 2 shows the information on health symptoms among family members. There were no significant differences between groups in health symptoms except for sore throat. The researchers then conducted a multiple logistic regression analysis. Residents who were exposed to the dumpsite were 1.9 times more likely to have a sore throat in the past one month compared with non-exposed residents (OR 1.88; 95% CI: 1.05, 3.38; *p*-value 0.031) when confounders were adjusted. The results are shown in Table 3.

### 3.2. Comparison of Self-Reported Diseases between Exposed and Non-Exposed Residents

The disease prevalence among family members diagnosed while residing in the study area is shown in Table 4. There were no significant differences except for pneumonia, hepatitis A, diabetes mellitus, hypertension, ischemic heart disease, and enuresis. When the researchers conducted multivariable analysis, only diabetes mellitus and hypertension yielded a significant association with dumpsite exposure. Residents who were exposed to the dumpsite were 2.8 times more likely to have diabetes mellitus (OR 2.84; 95% CI: 1.10, 7.30; *p*-value 0.021) and 2.6 times more likely to have hypertension (OR 2.56; 95% CI: 1.27, 5.13; *p*-value 0.006) than the non-exposed residents when confounders were adjusted (Table 5 and Table 6).

### 3.3. Comparison of Reproductive Health between Exposed and Non-Exposed Residents

Table 7 shows no significant differences between groups in reproductive health, except for the death of children aged less than five years old. There were more deaths of children under five in the exposed group compared to the unexposed group. However, there was no significant association between dumpsite exposure and child death in the multivariable analysis.

## 4. Discussion

Our study showed that residents living near the Sabak dumpsite had a significantly higher risk of having a sore throat, diabetes mellitus, and hypertension compared to residents in the control area. However, our study did not detect a significant association with other outcomes. These findings were supported by a number of community health surveys that investigated a wide range of health problems related to environmental exposure to a landfill [17]. An increased prevalence of self-reported health symptoms such as fatigue, sleepiness, and headache among residents living near waste sites has been reported [8]. Residents living near a landfill in South Africa reported poor air quality related to the landfill. Influenza-like illness, eye irritation, and body weakness were frequently reported by participants living closer to the landfill than those living far from the landfill [14]. A cross-sectional study was conducted among waste collector workers in Kota Bharu. It was reported that 75.0% of them had chronic respiratory symptoms (cough, phlegm, asphyxiate, and wheezing), 70.3% had dermatological symptoms (itchy and rashes), and 65.5% had gastrointestinal symptoms (nausea and diarrhea) [20]. However, the study was a descriptive study, so it did not control for potential confounders and did not have a comparison group.

The significant association between dumpsite exposure and diabetes and hypertension in our study was likely related to the presence of heavy metals in the dumpsite, such as arsenic, lead, cadmium, and mercury. Many industrial products, such as batteries and electrical equipment, contain heavy metals that can end up in the solid waste dumpsite. Heavy metals pose a risk to the health of persons exposed during collecting and handling by inhalation or ingestion or through skin contact. The impact on human health can also occur along the food chain, such as in vegetables grown or animals raised on contaminated soil. Heavy metals produce free radicals that disrupt intracellular homeostasis and damage lipids, proteins, enzymes, and DNA in the human body [21].

Long-term exposures, such as those experienced by people living for many years in areas with high concentrations of fine particles, have been linked to health problems such as reduced lung function, the development of chronic bronchitis, and even premature death. Short-term exposure to particles can worsen lung diseases, causing asthmatic attacks and acute bronchitis, and may also increase susceptibility to respiratory infections [22]. People who lived in a former landfill area in Helsinki were 1.63 times more likely to have asthma compared to the cohort of people living in similar rental apartments nearby but clearly outside the landfill [18]. Environmental monitoring at the Nant-y-Gwyddon landfill in South Wales has identified various emitted gases dominated by high levels of hydrogen sulphide, which caused significant complaints of headache, eye irritation, and sore throat among residents living within a 3 km radius from the landfill [23]. A study revealed that living near a landfill could reduce the function of the immune system and lead to an increased risk of infections due to the direct exposure to chemicals. The study found that children living near waste sites, whether landfills or contaminated bodies of water, were hospitalized more frequently due to acute respiratory infections. Children living near waste sites also had significantly increased rates of asthmatic attacks [24].

Our study did not report a significant association between dumpsite exposure and cancer, supported by a study conducted for the urban waste dumps in the municipality of São Paulo [25] and a study conducted in Great Britain [26]. A systematic review concluded that there was inadequate evidence to link landfills with the occurrence of cancer [12,19]. This was in contrast with a cohort study conducted in Finland that found pancreatic cancer and skin cancer were significantly more common among males who were exposed to the landfill with a longer number of years living in the area, but the causal association has not been confirmed [18].

Some of the substances in LFG can interfere with the development of embryos and fetuses. This can lead to infertility, intrauterine death, spontaneous abortion, low birth weight, and congenital anomalies. An ecological study reported that residents living within approximately 3 km of the Nant-y-Gwyddon landfill site in South Wales had a significant two-fold increase of maternal risk of having baby with a congenital abnormality [23]; this finding was supported by a review by Guisti [7], in contrast with a study by Gouveia and do Prado [25]. Golberg et al. reported that low birth weight was significantly elevated with the odd ratio of 1.2 among residents living within a 2 km radius from a municipal solid waste landfill in Montreal, Quebec [27]. However, our study failed to find a significant association between solid waste dumpsite exposure and reproductive history, as our study had a relatively small sample size and a low incidence of health events. These facts were supported by a systematic review by Mattiello et al. [19].

Our study used the WHO definition of landfill exposure, which defines exposure as within a 2 km radius from the landfill [17,26]. The distance of 1 to 2 km is conceptually supported by the WHO definition of landfill exposure, as transmission of chemicals and microbiological agents mainly through water and air pathways is presumed within a radius of 2 km [17]. The control zone was situated more than 2 km away and served as a reference zone. Therefore, our study has a low possibility of misclassification of exposure status. In addition, our study was a population-based study, with a survey representative of the community. Our comparison group in this study was similar in geographic, socio-cultural, and behavioral aspects as the exposed group, with the exception of the dumpsite exposure. We also took into consideration the internal validity of our study. A systematic review has shown a potential risk of bias when measuring the exposure, outcome, and confounding factors of any study on the health effects associated with the disposal of solid waste in landfills [19]. We controlled for the possible confounders of the study, such as age, smoking status, and duration of exposure by using multivariable statistical analysis.

### 4.1. Limitations of the Research

There are several limitations of this study. This is a cross-sectional study, measuring a mere association between dumpsite exposure and the health effects. The cause-effect relationship cannot be ascertained. Our study also relied on self-reported symptoms, which may be subject to bias. Neutra et al. commented that although self-reported symptoms may be subject to bias, they might be more sensitive indicators of exposure than diseases such as cancer with long latencies [28].

It was reported that the significant difference in the incidence of specific clinical effects between the two populations is usually small and the power of the investigation relies on the sample size [7]. Therefore, conducting an epidemiological study for a low-incidence disease needs adequate statistical power in order to avoid making false conclusions, requiring studying a large sample size of at least thousands of people in the exposure and control areas. Other limitations of our study were insufficient data on population mobility and the long latency period of some illnesses such as cancer. Many previous studies were generally found to have limitations such as exposure assessment and contamination, ecological level of analysis, and lack of information on confounding factors [12].

### 4.2. Policy Implications

Our study supported the findings of a systematic review that reported limited evidence of a relationship between landfill and health effects [12]. Landfills are the most common method for disposing of waste, and present a great challenge that local government, political leaders, and environmental departments must address. They have an effect on the environment, including the water, air, soil, landscape, and climate [5]. Integrated waste management practices are recommended, including recycling, organic waste management, energy recovery, and sanitary landfills. Given the limited space available for the landfill development and the environmental pollution that this may create; landfills cannot be the ultimate option for much longer. Technologies may be used to improve the treatment and disposal processes for solid waste. The commitment to support environmentally friendly activities such as recycling in promoting waste reduction as a key objective of the waste management policy also needs to be encouraged. The amount of waste generated continues to increase due to the population growth and development, and less than 5% of waste is being recycled. Thus, 3R (reduce, reuse, and recycle) should be practiced in solid waste management, to reduce dependence on the use of natural resources which are increasingly limited.

## 5. Conclusions

In conclusion, our study provides epidemiological evidence of the potential effect on human health of a dumpsite and serves as a basis for future research on the landfill impact in Malaysia. There were significant associations between exposure to the solid waste dumpsite and sore throat, diabetes mellitus, and hypertension. The close proximity to the open dumpsite is a risk factor, but we cannot say that the open dump is exactly the factor that causes the disease. Our results have implications to the residents that living near an open dumpsite is hazardous to health. We hope that this research will attract the attention of municipalities, district health officers, and nearby residents.

We suggest future research on larger samples that also include other municipal landfills. A cohort study monitoring the health status of residents will provide strong epidemiological evidence of the adverse effects of solid waste disposal. In addition, environmental monitoring and risk measurement will provide the scientific basis of the health effects.

## Figures and Tables

**Figure 1 ijerph-17-00311-f001:**
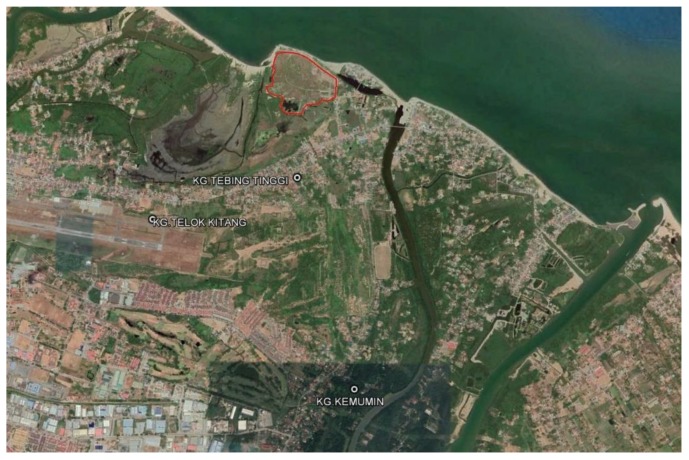
Satellite map of Sabak dumpsite, Kelantan, Malaysia.

**Figure 2 ijerph-17-00311-f002:**
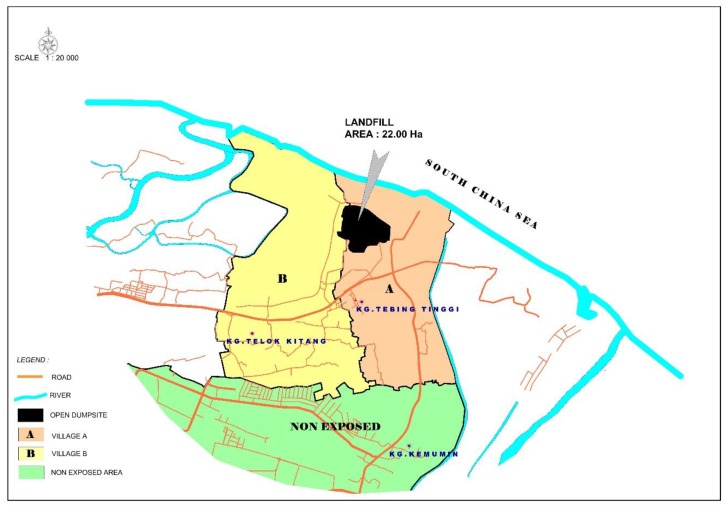
Map of Sabak dumpsite and the included villages in this study.

**Table 1 ijerph-17-00311-t001:** Socio-demographic comparison between residents in the exposed and non-exposed groups to the solid waste dumpsite in Sabak.

Socio-Demographic Characteristics	Frequency (%)/Mean (SD)		*p-*Value
	Exposed *n* = 170	Non-Exposed *n* = 119	
Age (year)	46.3 (13.5)	38.1 (9.8)	<0.001 _a_*
Sex			0.024 _b_*
Male	48 (28.2)	20 (16.8)	
Female	122 (71.8)	99 (83.2)	
Monthly household income (RM)	898 (912)	1028 (817)	0.213 _a_
Education level			0.002 _b_*
Primary school/less	64 (37.6)	22 (18.5)	
Lower secondary school	33 (19.4)	26 (21.8)	
Upper secondary school	57 (33.5)	49 (41.2)	
Above secondary school	16 (9.4)	22 (18.5)	
Occupation			0.277 _b_
Housewife	91 (53.5)	64 (53.8)	
Unemployed	17 (10.0)	6 (5.0)	
Employed	62 (36.5)	49 (41.2)	
Working in factory			0.021 _b_*
Yes	65 (2.9)	11 (9.2)	
No	165 (97.1)	108 (90.8)	
Scavenger at the dumpsite			0.095 _b_
Yes	7 (4.1)	1 (0.8)	
No	163 (95.4)	118 (99.2)	
Water Supply			<0.001 _b_*
Tube Well	6 (3.5)	3 (2.5)	
Dug Well	22 (12.9)	58 (48.7)	
Kelantan Water Company	142 (83.5)	58 (48.7)	
Smoking Status			0.095 _b_
Yes	25 (14.7)	10 (8.4)	
No	142 (83.5)	109 (91.6)	
Smoking among household members			0.763 _b_
Yes	67 (39.4)	49 (41.2)	
No	103 (60.6)	70 (58.8)	
Growing own vegetable			0.014 _b_*
Yes	28 (16.5)	34 (28.6)	
No	142 (83.5)	85 (71.4)	
Rearing own chicken			0.478 _b_
Yes	56 (32.9)	44 (37.0)	
No	114 (67.1)	75 (63.0)	
Duration of residence (year)	22.6 (18.9)	15.0 (12.0)	<0.001 _a_*
Distance from dumpsite (m)	450 (244)	3057 (374)	<0.001 _a_*

_a_ Independent *t* test, _b_ Chi square test, * statistically significant <0.05.

**Table 2 ijerph-17-00311-t002:** Comparison of health symptoms in past one month among family members of residents of the exposed and unexposed groups to the solid waste dumpsite in Sabak.

Health Symptoms	Frequency (%)		*p-*Value _a_
	Exposed *n* = 170	Non-Exposed *n* = 119	
Eye irritation			0.216
Yes	47 (27.6)	41 (34.5)	
No	123 (72.4)	78 (65.5)	
Skin rashes			0.444
Yes	69 (40.6)	43 (36.1)	
No	101 (59.4)	76 (63.9)	
Nasal irritation			0.878
Yes	50 (29.4)	36 (30.3)	
No	120 (70.6)	83 (69.7)	
Headache			0.192
Yes	70 (41.2)	40 (33.6)	
No	100 (58.8)	79 (66.4)	
Excessive tiredness			0.369
Yes	45 (26.5)	26 (21.8)	
No	125 (73.5)	93 (78.2)	
Excessive day sleepiness			0.656
Yes	38 (22.4)	24 (20.2)	
No	132 (77.6)	95 (79.8)	
Sore throat			0.041 *
Yes	51 (30.0)	23 (19.3)	
No	119 (70.0)	96 (80.7)	
Diarrhea			0.379
Yes	13 (7.6)	6 (5.0)	
No	157 (92.4)	113 (95.0)	
Stomachache			
Yes	17 (10.0)	6 (5.0)	0.125
No	153 (90.0)	113 (95.0)	

_a_ Chi square test, * statistically significant <0.05.

**Table 3 ijerph-17-00311-t003:** The association between dumpsite exposure and sore throat symptom.

Factors	b	Crude OR _a_ (95% CI)	Adjusted OR _b_ (95% CI)	*p-*Value
Exposure to dumpsite	0.632	1.789 (1.021, 3.135)	1.881 (1.048, 3.375)	0.031 *
Smoking	1.112	2.846 (1.376, 5.886)	3.04 (1.439, 6.421)	0.004 *
Vegetable grower	0.778	1.682 (0.914, 3.092)	2.177 (1.145, 4.139)	0.019 *

_a_ Simple logistic regression, _b_ Multiple logistic regression, * statistically significant <0.05.

**Table 4 ijerph-17-00311-t004:** Comparison of prevalence of diseases among family members between the exposed and unexposed groups to the solid waste dumpsite in Sabak.

Health Symptoms	Frequency (%)		*p-*Value
	Exposed *n* = 170	Non-Exposed *n* = 119	
Tuberculosis			0.146 _a_
Yes	4 (2.4)	0 (0.0)	
No	166 (97.6)	119 (100.0)	
Asthma			0.073 _b_
Yes	30 (17.6)	12 (10.1)	
No	140 (82.4)	107 (89.9)	
Pneumonia			0.044 _a_*
Yes	6 (3.5)	0 (0.0)	
No	164 (96.5)	119 (100.0)	
Typhoid fever			0.646 _a_
Yes	3 (1.8)	1 (0.8)	
No	167 (98.2)	118 (99.2)	
Cholera			1.000 _a_
Yes	1 (0.6)	0 (0.0)	
No	169 (99.4)	119 (100.0)	
Dengue fever			1.000 _a_
Yes	1 (0.6)	1 (0.8)	
No	169 (99.4)	118 (99.2)	
Hepatitis A			0.044 _a_*
Yes	6 (3.5)	0 (0.0)	
No	164 (96.5)	119 (100.0)	
Food poisoning			0.477 _a_
Yes	6 (3.5)	2 (1.7)	
No	164 (96.5)	117 (98.3)	
Diabetes mellitus			0.007 _b_*
Yes	26 (15.3)	6 (5.0)	
No	144 (84.7)	113 (95.0)	
Hypertension			<0.001 _b_*
Yes	50 (29.4)	13 (10.9)	
No	120 (70.6)	106 (89.1)	
Cancer			1.000 _a_
Yes	1 (0.6)	1 (0.8)	
No	169 (99.4)	118 (99.2)	
Ischemic heart disease			0.045 _b_*
Yes	14 (8.2)	3 (2.5)	
No	156 (91.8)	116 (97.5)	
Epilepsy			0.533 _a_
Yes	7 (4.1)	3 (2.5)	
No	163 (95.9)	116 (97.5)	
Enuresis among children			0.009 _b_*
Yes	10 (5.9)	18 (15.1)	
No	160 (94.1)	101 (84.9)	
Learning problem among children			0.126 _b_
Yes	8 (4.7)	11 (9.2)	
No	162 (95.3)	108 (90.8)	
Hyperactive children			0.176 _b_
Yes	12 (7.10)	4 (3.4)	
No	158 (92.9)	115 (96.6)	

_a_ Fisher’s exact test, _b_ Chi square test, * statistically significant <0.05.

**Table 5 ijerph-17-00311-t005:** The association between dumpsite exposure and diabetes mellitus.

Factors	b	Crude OR _a_ (95% CI)	Adjusted OR _b_ (95% CI)	*p-*Value
Exposure to dumpsite	0.021	3.400 (1.353, 8.543)	2.837 (1.103, 7.301)	0.021 *
Duration of exposure	0.021	1.027 (1.007, 1.047)	1.021 (1.001, 1.041)	0.043 *

_a_ Simple logistic regression, _b_ Multiple logistic regression, * statistically significant <0.05.

**Table 6 ijerph-17-00311-t006:** The association between dumpsite exposure and hypertension.

Factors	b	Crude OR _a_ (95% CI)	Adjusted OR _b_ (95% CI)	*p-*Value
Exposure to dumpsite	0.938	3.397 (1.749, 6.598)	2.555 (1.273, 5.130)	0.006 *
Age	0.035	1.047 (1.023, 1.071)	1.035 (1.011, 1.060)	0.003 *

_a_ Simple logistic regression, _b_ Multiple logistic regression, * statistically significant <0.05.

**Table 7 ijerph-17-00311-t007:** Comparison of reproductive health of residents in past 10 years between exposed and unexposed to the dumpsite.

Reproductive Health	Frequency (%)/Mean (SD)		*p-*Value
	Exposed *n* = 98	Non-Exposed *n* = 114	
Number of children	4.7 (2.8)	4.7 (2.9)	0.865 _a_
Average birth weight (kg)	3.06 (0.47)	3.11 (0.40)	0.405 _a_
Birth weight			0.570 _b_
Low	6 (6.1)	5 (4.4)	
Normal	92 (93.9)	109 (95.6)	
Abortion			0.573 _b_
Yes	12 (12.2)	17 (14.9)	
No	86 (87.8)	97 (85.1)	
Complication during pregnancy			0.722 _b_
Yes	10 (10.2)	10 (8.8)	
No	88 (89.8)	104 (91.2)	
Congenital malformed children			1.000 _c_
Yes	1 (1.0)	1 (0.9)	
No	97 (99.0)	113 (99.1)	
Death of children age <5 years			0.044 _c_*
Yes	4 (4.1)	0 (0.0)	
No	94 (95.9)	114 (100.0)	

_a_ Independent *t* test, _b_ Chi square test, _c_ Fisher’s exact test, * statistically significant <0.05.

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
