# Peer review of "Community Health Survey of Residents Living Near a Solid Waste Open Dumpsite in Sabak, Kelantan, Malaysia"

_ijerph, 2020, doi:10.3390/ijerph17010311_

Round 1

Reviewer 1 Report

The subjects investigated in your research are of interest to the journal. However, I believe that the statistical analysis made and the methods used for the analysis are not consistent and do not allow considering your research valid from a scientific point of view.

Both the structure of the paper and the research design are not appropriated. I would like also to underline that many results are just published in the literature about the negative impacts that open dumpsite and sanitary landfill cause to the population health and ecosystems.

You should better clarify what is the aim of your research, the new information provided that is not available in the scientific literature. For making that, you should before conducting a literature review about the subject, underline the holes left by the researchers in the field of solid waste final disposal and public health.

Finally, some technical errors are detectable within the text in the field of solid waste management, while only a few descriptions are provided about the current waste management system, making difficult to understand if the analysis can be biased by other factors. The sample is quite small and it is not justified by the analysis of the confidence interval and confidence level (how did you obtain it?). For these reasons, in my opinion, the article should be considerably improved and cannot be considered for publication in the current form.

While the topic of the research can be suited for the journal, the contents and the approaches can not be considered apporpriated for the scientific audience. The main issues are four:

The statistical analysis is not always appropriated or justified The references used are not suited in terms of time (you mostly cited paper dated 2006-2008) The results obtained cannot be accepted due to the lack of imperative information for excluding any bias in the research The methods and the outcomes are well known and just reported by the world health organization, as well as by other researches. I suggest reviewing the current scientific literature.

Therefore, I think that your paper is not suited for being published in a scientific journal.

I would like to provide you some specific comments for improving your research and paper:

I suggest adding the country name within the title

Abstract

L19 is it a village, town, city or megacity?

L24 This sentence is not clear. What do you mean with the mean duration?

L28 Please, define the novelty of the research and the main contribution to the scientific literature. The results are quite obvious without new relevant information.

Introduction

L34-40 Please, remove this part since it refers to the indications for writing the paper.

L63 K/Mg is not clear

This section lacks the introduction of the methods used in your research, as well as a brief description of the specific context where the research took place.

L84 This sentence is not clear. Which literature review did you implement? Which issue is not investigated? 

L86 I disagree with this statement. You should better revise the scientific literature about the subject of waste mismanagement in developing countries.

L87 There are many studies about landfill exposure and health effects. You should rethink the aim and scope of your research for highlighting the novelty of the research

You should better define the objectives and novelty of the research

Methods

L94 before, you should introduce the study area and the main issues detected in terms of solid waste management and environmental contamination.

L97 You should better describe the site

L99 What do you mean with prepared?

L101 What do you mean with "located within the defined zone"?

L104 Too general. Why and how did you choose this village? Where is it located? 

L106 randomly how? In function of the location, availability, list of participants...?

This first part is too general and should be removed, adding the information within specific sections.

L113 Where the waste come from?

L130 You should add a table with the list of questions. Why did you implement the Cronbach alpha? Which information would you like to collect? Which type of discussion among researchers did you implement?

L133 So, what would you like to investigate here?

L139 Was it implemented within the research?

L148 It is not clear how you implemented the random selection

L150 You should explain how and when the questionnaires were provided, and the type of answers (Likert scale, yes/no...)

L151 I think that 30-45 minutes is very much. You should justify this time for explaining how the interviews were carried out and who submitted it to the population. 

L160 You should better define how and why you implemented such specific statistical analysis

L164 Which model did you introduce?

Results

I suggest dividing the results section into sub-sections

L169 It is not clear how you choose the exposed ground and the non-exposed group

L171 What do you mean with the duration of residence?

L174 How did you measure the statistical significance? You should provide all the information about the statistical test used. Moreover, it is not clear how you choose the exposed group and the non-exposed group. How can you be sure of that? 

Table 1 

You should introduce the confidence interval for each question submitted

Are you sure that gender is statistically significant? The percentage obtained are very similar.

In my opinion, here there is an error in the statistical method used. Usually, the Chi-square is not used for analyzing the statistical significance of the difference between two groups, but the dependence that a question/value has with another. You should better explain why and how you used a statistical method rather than another.

L183 How can you be sure that the sore throat is generated due to the open dumpsite? You should better describe where the dumpsite is located. You cannot prove it only with this small sample and without explaining which activities are mainly implemented (if nearby the site other industrial activities or roads are located...). The sample is too small and the location is not well defined.

L186 What does 3.38 mean?

L187 Table 3 should be better explain. In my opinion, the results have not scientific significance. The health issue can be due to a different living standard rather than the presence of a sanitary landfill. In your research, this topic is not addressed and neither analyzed.

Table 2

The table should be uniformized. It can be useful to introduce the odds ratio for each health symptoms.

The difference in the statistical significance percentages is minimal and similar to other results. Are you sure about the statistical analysis implemented? The Chi-square test is not suited for comparing two different groups.

(Stars should be used only for defining the statistical significance. Please, revise scientific literature)

Table 3

It is not clear whit what the sore throat is regressed.

L197 How can you motivate this result? Heavy metals? Biological contamination? Within the text is not explained how the people are affected by the presence of the open dumpsite.

Discussion 

L215 In my opinion, this result is not demonstrated by your study due to too many biases in the research and the unclear use of statistical analysis. In my opinion, the research method should be reconsidered by the authors.

L218 These results were also detected by other studies.

L222 It depends on various factors. You should better describe such issue within the text

L224 Is it the case of the final disposal site?

L228 This data has no sense if no explanation is provided.

L229 These are landfill gases that are composed of CH4, CO2, among other components (< 1%). You should better explain the findings obtained by other studies.

L232 What do you mean with direct exposure

L236-241 Such results depend on the type of landfill, the type of exposure, the economic (health) conditions, among other factors. 

L253-256 This part should be presented within the method section.

L259 How can you say that your results are representative of the community? How did you calculate the confidence interval?

L262 This is not reported within the method section

L265 This part should be introduced within the method section for justifying the methods.

L270 This is my main concern about the findings of your research

L276 In my opinion, the reference is too old for justifying your study. Many other studies are available in the literature. I suggest improving your literature review.

L281 You introduce sanitary landfill and open dumping in the same manner. They are not the same systems and you should specify it in order to avoid misleading information.

The conclusions section should be improved. The main findings, pros, and cons of the approach, future developments and indications provided by the research should be provided. The novelty of the research is not clearly stated neither in the conclusion section. The information provided is not new for the scientific audience. You should provide more insights into the usefulness of your research.

References are too old. They should be improved in number, quality, and age. I suggest using literature not older than 10 years.

Author Response

Thank you for the reviewer’s comments. We have addressed the comments point by point as below. For your information, this paper had been English proofed [see the attachment certificate].

Response to Reviewer 1 Comments

I suggest adding the country name within the title

We add Malaysia in the title.

Abstract

L19 is it a village, town, city or megacity?

Sabak is a suburban.

L24 This sentence is not clear. What do you mean with the mean duration?

This means the average duration time of residence.

L28 Please, define the novelty of the research and the main contribution to the scientific literature. The results are quite obvious without new relevant information.

We think our study renew support of the previous literature and is a new finding in Malaysia. These findings provide an evidence for the policy maker to reduce production of solid waste and using sanitary methods of waste disposal.

Introduction

L34-40 Please, remove this part since it refers to the indications for writing the paper.

We are sorry & removed the paragraph.

L63 K/Mg is not clear

Potassium/Magnesium ratio

This section lacks the introduction of the methods used in your research, as well as a brief description of the specific context where the research took place.

I think our method is not outstanding that needs review in the introduction. Most manuscripts do not review their method in the introduction unless it is exceptional or a new method. This study is a cross-sectional in design.

L84 This sentence is not clear. Which literature review did you implement? Which issue is not investigated? 

Our literature search did not find many research conducted in Malaysia. Most of the studies were review papers.

L86 I disagree with this statement. You should better revise the scientific literature about the subject of waste mismanagement in developing countries.

We are citing this statement from Porta et al who did a meta-analysis on the association between waste management and health effects.

L87 There are many studies about landfill exposure and health effects. You should rethink the aim and scope of your research for highlighting the novelty of the research. You should better define the objectives and novelty of the research

We are sorry that this study was done and we could not change the aim of study.  There is no previous research like this study in Malaysia. We think our study is comprehensive that include health symptoms, illnesses, diseases and maternal and child health.

Methods

L94 before, you should introduce the study area and the main issues detected in terms of solid waste management and environmental contamination.

We have introduced the study area in 2.1 L111-128

L97 You should better describe the site

We have included two new maps of the area

L99 What do you mean with prepared?

We changed the word to “provided”

L101 What do you mean with "located within the defined zone"?

We meant that it is located within 1km radius from the landfill

L104 Too general. Why and how did you choose this village? Where is it located?

One village was chosen as non-exposed area with the distance more than 3km from the landfill. The village is right next to the exposed villages and chosen because we want to control the similarity of other environmental exposures other than the landfill.

L106 randomly how? In function of the location, availability, list of participants...?

We used systematic sampling to select consecutive second houses in each village.  The eligibility criteria include: (i) permanent resident of the area for at least one year and after the opening of the landfill, (ii) age 20 years and above, and (iii) head of the family or the next of kin. Residents were excluded if they had a mental problem, were not at home during visits, or had resided in a landfill area prior to residing in the control area.

This first part is too general and should be removed, adding the information within specific sections.

We are sorry that we did not understand the meaning of this. Which part? Please give suggestions.

L113 Where the waste come from?

It is written in L128. The waste was collected from the Kota Bharu area, the capital city and the most populated district of Kelantan state.

L130 You should add a table with the list of questions. Why did you implement the Cronbach alpha? Which information would you like to collect? Which type of discussion among researchers did you implement?

We researchers discussed on the list of questions. The list of questions / information was as in the tables 1, 2, 4 & 7. We ran Cronbach alpha of SPSS for the internal consistency of the questions.

L133 So, what would you like to investigate here?

Section A was the socio-demographic data

L139 Was it implemented within the research?

Yes as in table 4

L148 It is not clear how you implemented the random selection

It is explained in L117.

L150 You should explain how and when the questionnaires were provided, and the type of answers (Likert scale, yes/no...)

We have explained in detail L142-159. We included the questionnaires as supplementary. Most of the questions have Yes/No response. See the tables for the responses.

L151 I think that 30-45 minutes is very much. You should justify this time for explaining how the interviews were carried out and who submitted it to the population. 

We have added more explanation. We interviewed the residents one by one using the questionnaires.

L160 You should better define how and why you implemented such specific statistical analysis

A chi square test or Fisher’s exact test was used to compare the difference in proportions, while t test was used to compare the means difference of numerical variables.

We have added more explanations.

L164 Which model did you introduce?

The final model was selected that include landfill exposure, the best fitting and the simplest model possible; describing the association between the landfill exposure and other independent variables and the outcome such as health symptoms, diseases or reproductive health.

We follow the steps as in Hosmer Lemeshow. Applied Logistic Regression, Second Edition. 2000.

Results

I suggest dividing the results section into sub-sections

We add sub-sections

L169 It is not clear how you choose the exposed ground and the non-exposed group

We have explained earlier in the methods L99-108

L171 What do you mean with the duration of residence?

It means how long they had been staying in the area.

L174 How did you measure the statistical significance? You should provide all the information about the statistical test used. Moreover, it is not clear how you choose the exposed group and the non-exposed group. How can you be sure of that? 

The statistical significance was calculated by the SPSS software in each of the appropriate statistical analysis. We provide the detail statistical test in L169-185. We have explained on the selection of the exposed and non-exposed group in the methods L99-108. It is based on the radial distance from the landfill.

Table 1 

You should introduce the confidence interval for each question submitted

Presenting CI for each of the variable would be too lengthy and is usually not done by many researchers. We only provide CI of estimation for the final results.

Are you sure that gender is statistically significant? The percentage obtained are very similar.

Yes, we are sure because we re-analysed the variable and showed the same p value.

In my opinion, here there is an error in the statistical method used. Usually, the Chi-square is not used for analyzing the statistical significance of the difference between two groups, but the dependence that a question/value has with another. You should better explain why and how you used a statistical method rather than another.

“Chi-square is not used for analyzing the statistical significance of the difference between two groups”. We agree with the reviewer statement. It is not used to analyse statistical significance but test the difference of proportions between two groups. We added more sentences L172-174.

L183 How can you be sure that the sore throat is generated due to the open dumpsite? You should better describe where the dumpsite is located. You cannot prove it only with this small sample and without explaining which activities are mainly implemented (if nearby the site other industrial activities or roads are located...). The sample is too small and the location is not well defined.

We agree that we cannot confirm that the sore throat is caused by the landfill. We already discussed about the causal relationship L291. Our study only found that there was an association between sore throat and landfill exposure after smoking was controlled. While other environmental exposure was similar for both groups such as nearby the site other industrial activities or roads are located. These were controlled during the selection of the villages. Yes, we agree that the sample is not large but adequate. The location is well defined as shown in the map.

L186 What does 3.38 mean?

3.38 is the upper limit of the 95% confidence interval. With the adjusted odds ratio 1.88, 3.88 is relatively not wide, meaning that the precision of estimation is good. Therefore, we would say that the power of the study and the sample size are adequate.

L187 Table 3 should be better explain. In my opinion, the results have not scientific significance. The health issue can be due to a different living standard rather than the presence of a sanitary landfill. In your research, this topic is not addressed and neither analyzed.

Yes, we agree that the health issues could be due to different living standard, rather than the presence of unsanitary landfill. However, there was non-significant difference between the exposed and non-exposed groups in term of household income, see table 1.

Table 2

The table should be uniformized. It can be useful to introduce the odds ratio for each health symptoms.

The difference in the statistical significance percentages is minimal and similar to other results. Are you sure about the statistical analysis implemented? The Chi-square test is not suited for comparing two different groups.

(Stars should be used only for defining the statistical significance. Please, revise scientific literature)

There are several ways to present scientific results.  This table shows the association between health symptoms and landfill exposure at univariable level. We decided not to include the odds ratio because these are not the final results, unlike the results of multivariable analysis.

We do not agree with the reviewer regarding the statistical test.  To our best knowledge, statistical analyses applied in this study were appropriate to answer the research questions.  Pearson Chi-square test and Fisher's exact test are appropriate for comparing proportions between or among two or more independent groups. In our study, they were applied to compare proportions of categorical variables between exposed and non-exposed groups and we are confident that tests applied were appropriate. The results were adequate to be presented with Odds Ratio, 95 % CI and p-values in all tables. It was appropriate to have applied binary logistic regressions to answer the associated of landfill exposure and the outcome (sore throat, diabetes and hypertension respectively) at multivariable level.

Most of our references were using chi-square as well as logistic regression for their statistical analysis.

Table 3

It is not clear whit what the sore throat is regressed.

Table 3 shows that final model of multiple logistic regression for the sore throat. Sore throat is the binary dependent variable. The selected independent variables are landfill exposure, smoking and vegetable grower.

L197 How can you motivate this result? Heavy metals? Biological contamination? Within the text is not explained how the people are affected by the presence of the open dumpsite.

We have included new sentences that relate these results with waste exposure

Discussion 

L215 In my opinion, this result is not demonstrated by your study due to too many biases in the research and the unclear use of statistical analysis. In my opinion, the research method should be reconsidered by the authors.

We control biases and confounders during selection of areas and in multiple logistic regression. We have explained about the statistical test. We have discussed the weakness of research methods in the study’s limitation.

L218 These results were also detected by other studies.

Yes, we included more references.

L222 It depends on various factors. You should better describe such issue within the text

Please explain what do you mean? And give suggestion.

L224 Is it the case of the final disposal site?

What do you mean? Please provide suggestion.

L228 This data has no sense if no explanation is provided.

L229 These are landfill gases that are composed of CH4, CO2, among other components (< 1%). You should better explain the findings obtained by other studies.

High levels of hydrogen sulphide may cause headache, eyes irritation and sore throat among residents living within 3 km radius from the landfill. Hydrogen sulphide is a frequent occupational and environmental hazard. It’s health effects depends on the concentration level. Moderate concentration may cause nausea, headache and sore throat, while higher dose may cause eye damage and lung irritation.

L232 What do you mean with direct exposure

This mean straight contact of chemicals to humans either by inhalation, ingestion or skin contact.

L236-241 Such results depend on the type of landfill, the type of exposure, the economic (health) conditions, among other factors. 

Yes, we agreed.

L253-256 This part should be presented within the method section.

We already mentioned about the WHO criteria in the methods.

L259 How can you say that your results are representative of the community? How did you calculate the confidence interval?

Our results are representative of the community because we used systematic sampling method. We calculated the confidence interval by using SPSS software.

L262 This is not reported within the method section

We mention it L106-109.

L265 This part should be introduced within the method section for justifying the methods.

We think the point is more appropriate in the discussion, instead of in methods.

L270 This is my main concern about the findings of your research

Yes, we are aware of our study weakness.

L281 You introduce sanitary landfill and open dumping in the same manner. They are not the same systems and you should specify it in order to avoid misleading information.

We could not find & understand your point.

The conclusions section should be improved. The main findings, pros, and cons of the approach, future developments and indications provided by the research should be provided. The novelty of the research is not clearly stated neither in the conclusion section. The information provided is not new for the scientific audience. You should provide more insights into the usefulness of your research.

We have added few sentences.

L276 In my opinion, the reference is too old for justifying your study. Many other studies are available in the literature. I suggest improving your literature review.

References are too old. They should be improved in number, quality, and age. I suggest using literature not older than 10 years

We added more recent references.

Reviewer 2 Report

The topic of study is of significant interest for readers of IJERPH Journal. However, in the present format, the paper could be improved.

The stream literature review is incomplete and could be developed. Particularly, more references (5 to 8) about the findings of other researches could be cited. Please include the map of the area (City and concerned zone). This could give the reader a clear idea about the situation. Please submit the questionnaire file as supplementary material to help similar researches. Do you have any existent information of realized studies about the air and water table quality in the study area? Line 263: You mentioned, “This is a cross-sectional study, measuring a mere association between landfill exposure and health effect. The cause-effect relationship cannot be ascertained.” Please mention in the introduction what other researches did to verify the cause-effect relationship? May be the comparison with the cause – effect on waste pickers working on the landfill (0 Km) could help you (see comment 9) Line 281: Is it better to write “energy recovery” than “Incineration”? The discussion part would need to have more references to the theory referred to in the theoretical background (Introduction section), to indicate how the results respond to the research questions and can be explained. References presentation (different size, spaces, etc). Please give more attention to the references. Is there any waste picking activities in the Landfill? Is it possible to compare with the health situation of these people spending 4 to 8 hours in the landfill? It would help to confirm your results.

Author Response

Response to Reviewer 2 Comments

The stream literature review is incomplete and could be developed.

We added some literature review.

Please include the map of the area (City and concerned zone). This could give the reader a clear idea about the situation.

We have attached two maps of the area.

Please submit the questionnaire file as supplementary material to help similar researches.

We enclosed the questionnaires.

Do you have any existent information of realized studies about the air and water table quality in the study area?

We are sorry, there is no such study ever done in the area.

Line 263: You mentioned, “This is a cross-sectional study, measuring a mere association between landfill exposure and health effect. The cause-effect relationship cannot be ascertained.” Please mention in the introduction what other researches did to verify the cause-effect relationship? May be the comparison with the cause – effect on waste pickers working on the landfill (0 Km) could help you (see comment 9) Is there any waste picking activities in the Landfill? Is it possible to compare with the health situation of these people spending 4 to 8 hours in the landfill? It would help to confirm your results. 

The possible method to ascertain the cause-effect relationship is by conducting a cohort study. Cohort study will require a long follow-up on the participants, thus is not possible. There are many waste collectors in the area. We found a study among waste collector workers; Aminuddin that reported the health symptoms’ prevalence. 

Line 281: Is it better to write “energy recovery” than “Incineration”?

Yes, we changed the word

The discussion part would need to have more references to the theory referred to in the theoretical background (Introduction section), to indicate how the results respond to the research questions and can be explained.

We have added more references

References presentation (different size, spaces, etc). Please give more attention to the references.

It is done.

Round 2

Reviewer 1 Report

Dear author,

I think that you addressed appropriately most of the issues that I found during the first revision. However, there are still flaws that should be addressed for improving the quality of the paper and its contents. I hope that the following advises could help you in improving the text.

Abstract

You should add a statement of novelty, such as "It has never been found that a specific ... was statistically significant with ... " or "the methods used are new in terms of...", or "the case study and the discussion introduced is novel since...". You should not focus on the novelty of the research in Malaysia. This information should be added also within the introduction section. If not, the article has no merit for being published in an international scientific journal. 

L35-36 I suggest supporting these sentences with further scientific citations. IJERPH published some papers about the subject, which you can use as references. 

L80 You should better define "waste processing". Do you mean collection, recycling, final disposal, incineration...?

L81 This review is quite old (10 years ago!). I suggest considering recent articles and studies. The same is valid for the documents provided by the WHO.

L89 I suggest adding a reference here, specifying the WHO criteria (as you reported within the discussion section)

Figures 1 and 2 should be improved in quality. Moreover, I suggest defining the area for the analysis of the exposed sample and of the non-exposed one.

L111-116 please, remove italics

L116 move the title of the subsection below the section in italics

L121 Therefore, I suggest changing the word landfill with 'open dumpsite', or 'uncontrolled final disposal site'.

L132-134 I suggest removing this sentence

L137 Please, can you better define the term "discussion"? Is it a meeting with local stakeholders, did you interview local experts, is it simply a literature review of previous studies, did you implement a pre-test?

L138 You should explain why you implemented the Cronbach alpha. How did you measure it? For what purpose? What is the group of questions assessed? Are you sure that its use is appropriated here?

L154 I suggest avoiding this form. It is better "The residents were recruited by door-to-door interviews", and it is valid within the whole text.

L165-167 I suggest explaining the function of the chi-square as you mentioned within the answers "Pearson Chi-square test and Fisher's exact test are appropriate for comparing proportions between or among two or more independent groups. In our study, they were applied to compare proportions of categorical variables between exposed and non-exposed groups ". You should underline it also within the conclusions section in order to avoid misunderstanding in terms of research findings. Indeed, You can say that the proximity to the open dumpsite is a factor of risk, but you cannot say that the open dumpsite is exactly the factor that causes the illnesses. 

L170 I suggest changing the example or motivating why age is an important factor that should be evaluated

L175 explain which independent variables

L187 I suggest adding some more explanation about the results obtained, justifying the findings (i.e. the standard of leaving near the final disposal site is of lower quality, a lower economic situation...). Two photos of the final disposal site can be added within the method section in order to provide evidence about the current situation.

Tables: I understand that you disagree, but I suggest again changing the stars with letters (ex. a, b), adding the * when the result is statistically significant. It is more appropriated and allows a better reading of the table.

L247 This sentence should be better supported. It is not a result of your research. Please, see the comments concerning the use of the chi-square test. I suggest support these statements with references, or I suggest avoiding it.

L249 Where is it located? It should be reported within the figures 1-2 since it can be another factor of risk that should be included in the study

L259 I suggest removing this part since it is not clear. Or you can deepen it.

L324 I suggest adding more data obtained and implications. Who can be interested in your findings? What is the novelty of the research - see the previous comment -? It should be also stated here.

Author Response

Abstract

You should add a statement of novelty, such as "It has never been found that a specific ... was statistically significant with ... " or "the methods used are new in terms of...", or "the case study and the discussion introduced is novel since...". You should not focus on the novelty of the research in Malaysia. This information should be added also within the introduction section. If not, the article has no merit for being published in an international scientific journal. 

L 21-22. It has never been found that research linked to solid waste dumping exposure with health symptoms, chronic diseases and the child and maternal health together in one study.

L35-36 I suggest supporting these sentences with further scientific citations. IJERPH published some papers about the subject, which you can use as references. 

We added Yukalang, N.; Clarke, B.; Ross, K. Solid waste management solutions for a rapidly urbanizing area in Thailand: recommendations based on stakeholder input. Int. J. Environ. Res. Public Health 2018, 15, 1302; doi:10.3390/ijerph15071302

L80 You should better define "waste processing". Do you mean collection, recycling, final disposal, incineration...?

It was mentioned in Porta et al that waste processing was referred to mainly landfills and incineration.

L81 This review is quite old (10 years ago!). I suggest considering recent articles and studies. The same is valid for the documents provided by the WHO.

We searched many times for other quality articles in recent publication to replace Porta et al, and WHO but we could not find it. We hope that reviewer can suggest recent articles.

L89 I suggest adding a reference here, specifying the WHO criteria (as you reported within the discussion section)

Done

Figures 1 and 2 should be improved in quality. Moreover, I suggest defining the area for the analysis of the exposed sample and of the non-exposed one.

We redefined the areas. The figures looked poor in quality because we paste the figure in the Microsoft word. However, we also enclosed separately the high quality figures. We hope they are acceptable.

L111-116 please, remove italics

Done

L116 move the title of the subsection below the section in italics

Done

L121 Therefore, I suggest changing the word landfill with 'open dumpsite', or 'uncontrolled final disposal site'.

We changed to open dumpsite

L132-134 I suggest removing this sentence

It does not control the adverse environmental impacts of landfills, such as groundwater contamination by leachate and landfill gas (LFG) emissions. Since 2001, the waste has been openly dropped and not filled up.

Do you mean this sentences? Why should they be deleted?

L137 Please, can you better define the term "discussion"? Is it a meeting with local stakeholders, did you interview local experts, is it simply a literature review of previous studies, did you implement a pre-test?

I think you mean L139. Yes, we forgot mentioning that we did meet with persons in charge from the municipal council and health clinic. Thank you for questioning this.

L138 You should explain why you implemented the Cronbach alpha. How did you measure it? For what purpose? What is the group of questions assessed? Are you sure that its use is appropriated here?

We deleted the sentences.

L154 I suggest avoiding this form. It is better "The residents were recruited by door-to-door interviews", and it is valid within the whole text.

Ok, we replaced the sentence.

L165-167 I suggest explaining the function of the chi-square as you mentioned within the answers "Pearson Chi-square test and Fisher's exact test are appropriate for comparing proportions between or among two or more independent groups. In our study, they were applied to compare proportions of categorical variables between exposed and non-exposed groups ". You should underline it also within the conclusions section in order to avoid misunderstanding in terms of research findings. Indeed, You can say that the proximity to the open dumpsite is a factor of risk, but you cannot say that the open dumpsite is exactly the factor that causes the illnesses. 

Thank you for your suggestions. We follow the suggestions given.

L170 I suggest changing the example or motivating why age is an important factor that should be evaluated

We removed age.

L175 explain which independent variables

We included all the variables.

L187 I suggest adding some more explanation about the results obtained, justifying the findings (i.e. the standard of leaving near the final disposal site is of lower quality, a lower economic situation...). Two photos of the final disposal site can be added within the method section in order to provide evidence about the current situation.

Thank you for the good suggestion. We added more explanations. However, we were unable to take photos due to the rainy season.

Tables: I understand that you disagree, but I suggest again changing the stars with letters (ex. a, b), adding the * when the result is statistically significant. It is more appropriated and allows a better reading of the table.

Done

L247 This sentence should be better supported. It is not a result of your research. Please, see the comments concerning the use of the chi-square test. I suggest support these statements with references, or I suggest avoiding it.

Ok we deleted that sentence.

L249 Where is it located? It should be reported within the figures 1-2 since it can be another factor of risk that should be included in the study

I think you misunderstood the statement. In previous review, you asked us to give explanation on the association. We did not say that there is a heavy metal dump area. We say that solid waste most likely contains heavy metals from electrical equipment and batteries that may cause chronic diseases like diabetes and hypertension. We added a reference.

L259 I suggest removing this part since it is not clear. Or you can deepen it.

We are sorry not to know which sentences? According to my document L259 begins with “Pukkala et al…” do you want it removed? We have rephrased it.

L324 I suggest adding more data obtained and implications. Who can be interested in your findings? What is the novelty of the research - see the previous comment -? It should be also stated here.

We added the stakeholders in the conclusion.

Reviewer 2 Report

The article has been considerably developed.  Please translate the questionnaire to English. 

Author Response

We have translated the questionnaire. See the supplementary file. Thank you.

Round 3

Reviewer 1 Report

Dear authors,

In my opinion, you improved considerably your paper.

I suggest adjusting it with some minor revisions.

Answering to your question, I suggest the following publications (They can be added to your paper in order to improve the references list, and you can take them as reference for future research):

1. Mattiello, A., Chiodini, P., Bianco, E., Forgione, N., Flammia, I., Gallo, C., ... & Panico, S. (2013). Health effects associated with the disposal of solid waste in landfills and incinerators in populations living in surrounding areas: a systematic review. International journal of public health, 58(5), 725-735.

2. World Health Organization. Waste and human health: evidence and needs. WHO Meeting Report: 5-6 November 2015. Bonn, Germany. Copenhagen: WHO Regional Office for Europe; 2016.

3. Ferronato, N., & Torretta, V. (2019). Waste mismanagement in developing countries: A review of global issues. International journal of environmental research and public health, 16(6), 1060.

4 Fazzo, L., Minichilli, F., Santoro, M., Ceccarini, A., Della Seta, M., Bianchi, F., ... & Martuzzi, M. (2017). Hazardous waste and health impact: a systematic review of the scientific literature. Environmental Health, 16(1), 107.

L190. 95% confidence level (It is not a confidence interval).  Rephrase as: "...adjusted odds ratio and confidence interval (95% confidence level). The level of statistical significance was set at p-value < 0.05". I found this mistake also in Tables (i.e. Table 3. Please, correct as (CI, 95% confidence level)

Tables: Letters can be introduced as subscript (i.e.  < 0.001a*)

I suggest dividing the discussion section into subsections, underlining the "policy implications" and "limits of the research".

Hope that my revisions helped you to improve your paper. Good luck with your future research.

Regards,

Reviewer
